# Development of a VLP-Based Vaccine Displaying an xCT Extracellular Domain for the Treatment of Metastatic Breast Cancer

**DOI:** 10.3390/cancers12061492

**Published:** 2020-06-08

**Authors:** Valeria Rolih, Jerri Caldeira, Elisabetta Bolli, Ahmad Salameh, Laura Conti, Giuseppina Barutello, Federica Riccardo, Jolanda Magri, Alessia Lamolinara, Karla Parra, Paloma Valenzuela, Giulio Francia, Manuela Iezzi, Federica Pericle, Federica Cavallo

**Affiliations:** 1Department of Molecular Biotechnology and Health Sciences, Molecular Biotechnology Center, University of Torino, 10126 Torino, Italy; valeria.rolih@gmail.com (V.R.); elisabetta.bolli@unito.it (E.B.); laura.conti@unito.it (L.C.); giuseppina.barutello@unito.it (G.B.); federica.riccardo@unito.it (F.R.); jolanda.magri@unito.it (J.M.); 2AgilVax Inc., Albuquerque, NM 87110, USA; jcaldeira@agilvax.com (J.C.); nsalameh@agilvax.com (A.S.); federica_pericle@outlook.com (F.P.); 3Department of Medicine and Aging Sciences, Center for Advanced Studies and Technologies (CAST), G. d’Annunzio University of Chieti-Pescara, 66100 Chieti, Italy; alessia.lamolinara@gmail.com (A.L.); m.iezzi@unich.it (M.I.); 4Border Biomedical Research Center, University of Texas at El Paso, El Paso, TX 79968, USA; Karla.Parra@utsouthwestern.edu (K.P.); valenzuelapa@nih.gov (P.V.); gfrancia@utep.edu (G.F.)

**Keywords:** xCT, anti-tumor vaccine, metastasis, breast cancer

## Abstract

Metastatic breast cancer (MBC) is the leading cause of cancer death in women due to recurrence and resistance to conventional therapies. Thus, MBC represents an important unmet clinical need for new treatments. In this paper we generated a virus-like particle (VLP)-based vaccine (AX09) to inhibit de novo metastasis formation and ultimately prolong the survival of patients with MBC. To this aim, we engineered the bacteriophage MS2 VLP to display an extracellular loop of xCT, a promising therapeutic target involved in tumor progression and metastasis formation. Elevated levels of this protein are observed in a high percentage of invasive mammary ductal tumors including triple negative breast cancer (TNBC) and correlate with poor overall survival. Moreover, xCT expression is restricted to only a few normal cell types. Here, we tested AX09 in several MBC mouse models and showed that it was well-tolerated and elicited a strong antibody response against xCT. This antibody-based response resulted in the inhibition of xCT’s function in vitro and reduced metastasis formation in vivo. Thus, AX09 represents a promising novel approach for MBC, and it is currently advancing to clinical development.

## 1. Introduction

Metastatic breast cancer (MBC), considering both metastases discovered at diagnosis (de novo) and those arising later (recurrences), represents the most severe form of this neoplastic disease [1]. Actually, the approaches available to treat MBC are chemotherapy, radiotherapy, endocrine treatments, surgery and monoclonal antibody immunotherapies [2]. All these strategies attempt to prolong survival and ameliorate the quality of patient life, but actually, none of them is able to cure the disease [3]. Indeed, breast cancer currently represents the second cause of cancer death [4]. Thus, there is an urgent unmet need for new effective therapies for MBC.

Breast cancer cells easily spread from the primary tumor to distant organs such as the lung, liver, bone and brain, leading to metastasis formation [5]. Since these circulating and disseminated cells have features similar to stem cells, they are referred to as breast cancer stem cells (BCSCs) [6]. BCSCs are resistant to chemotherapy and radiotherapy, thanks to the up-regulation of detoxification enzymes and drug efflux molecules, to an increased DNA repair activity [7] and to immune-evasive features [8]. We have previously demonstrated [9] that BCSCs overexpress xCT, the substrate-specific subunit of the heterodimeric amino-acid transport system x_c-_. xCT is a membrane protein with six extracellular domains (ECD1–6) [10] that mediates the export of intracellular glutamate in exchange for extracellular cystine [11], required for glutathione (GSH) production and therefore for radical oxygen species (ROS) detoxification [11,12].

Indeed, xCT protein is overexpressed in different tumor types, including Her2-positive and triple-negative MBC, and correlates with poor prognosis [13,14,15,16,17,18,19,20]. In tumor cells, xCT interacts with different molecules such as Mucin-1 [21], epithelial growth factor (EGF) receptor [22] and CD44v [23] supporting tumor growth, metastasis formation and drug resistance [24,25]. For example, the binding with CD44v promotes xCT stabilization at the plasma membrane, which results in an increased GSH synthesis and protects CSCs from oxidative stress [24]. xCT is not only involved in the GSH synthesis/ROS detoxification pathway but also protects cells from ferroptosis [26] and regulates glucose and glutamine metabolism [27].

Moreover, its expression is constitutively low and restricted to a few normal cell subpopulations such as astrocytes, microglia, other subsets of myeloid cells and activated T cells [28,29,30,31]. Recently, Arensman and colleagues [32] demonstrated that xCT has a causal role in tumor progression, while it is dispensable for T cell activation in vivo. They demonstrated that the deletion of xCT in murine colon and pancreatic cancer cell lines reduces tumor growth, while the absence of xCT does not affect the proliferative capacity of T cells [32]. These results are in line with what was reported by Chen and colleagues, demonstrating that xCT inhibition reduces tumor growth and metastasis formation in an esophageal cancer mouse model [33]. All these data support our hypothesis reported by Lanzardo and colleagues that xCT is a promising target for MBC treatment [9].

We have recently shown that anti-cancer vaccines targeting xCT elicit antibodies against xCT with anti-tumor activity in mouse models of breast cancer [9,34,35,36]. Here, we focus on a new vaccine (AX09) based on a virus-like particle (VLP) displaying the third extracellular domain (ECD3) of the xCT protein. To generate the AX09, we inserted the human ECD3 peptide into the AB-loop (surface coat protein) of the bacteriophage MS2 coat protein. When expressed in *E. coli*, this recombinant protein self-assembles into a VLP [37]. Because of its multivalence and small size, the VLP is strongly immunogenic, as we have extensively discussed in [38]. The presentation of epitopes in a dense repetitive array is highly stimulatory to B-cells, and the nano-size of the particle enhances its uptake by antigen-presenting cells [39]. In fact, the platform is so immunogenic that it can break B cell tolerance and elicit antibodies against self-antigens [40]. VLP-based vaccines are already licensed for infectious diseases and cancer-correlated pathogens [41,42], thus AX09 presents a high translational value. Of note, AX09 represents a novelty in the VLP family of vaccines, as no VLP targeting cancer antigens has been approved for use in the clinic so far [38,41,42].

Moreover, the AX09 vaccine is an innovative therapeutic approach because it targets the CSCs responsible for recurrence and metastasis formation; the treatment directly stimulates the patient’s immune system and should have fewer side effects compared to conventional therapy.

In this paper, we test the AX09 vaccine in several MBC mouse models, demonstrating that it is safe, induces a high titer of anti-xCT antibody and affects metastasis progression. In addition, we set up a pipeline for the GMP (Good Manufacturing Practices) production of AX09. These data support and encourage clinical trials for MBC patients.

## 2. Results

### 2.1. AX09 VLP Generation

The AX09 vaccine was generated by the insertion of human ECD3 into the AB-loop of the MS2 coat protein in a pDSP62 plasmid. The recombinant protein was expressed in *E. coli*, where it self-assembled into a VLP with ECD3 displayed multivalently on its surface [36] (Figure 1A). The particle is relatively stable, denaturing only above 50 °C (Figure 1B). The time course of denaturation by incubating the AX09 for different times at a fixed temperature of 55 °C was also evaluated (Figure 1C). These results show that 40% of the AX09 remains intact after 5 min at 55 °C. AX09 is also resistant to cycles of freezing and thawing (Figure 1D) and survives at least a month of storage at low temperatures (Figure 1E).

These data suggest that AX09 storage does not require any stabilizing protein cocktail, cryoprotectants or protease inhibitors.

### 2.2. Optimization of AX09 Immunization Protocol and Evaluation of the Antibody Response

To define the optimal dose regimen for AX09 administration, we immunized BALB/c mice with 2.5, 5, 10 and 20 µg of AX09 into the right caudal thigh muscle at Day 0 and 21 (two injections at 3 week intervals) and at Day 0, 14 and 28 (three injections at 2 week intervals). One week after the last immunization, sera were collected and evaluated by ELISA for their ability to bind synthetic human ECD3 peptide. As shown in Figure 2A, sera from the mice immunized with AX09 were positive in the ELISA at all doses (2.5, 5, 10 and 20 μg) and in both regimens (two versus three vaccinations). The 10 μg dose with three vaccinations at 2 week intervals gave the best antibody response. We selected this dose to further characterize the vaccine-induced humoral response, exploiting an ELISA assay for IgG subclasses. As reported in Figure 2B, AX09 induced high levels of anti-xCT IgG1 and IgG2a that, through FcγR-binding, could promote efficient anti-cancer mechanisms [43], which, in the case of IgG2a, include antibody-dependent cellular cytotoxicity (ADCC) and complement-dependent cytotoxicity (CDC).

BALB/c mice are biased towards a T helper (Th)2 response [44] and thus prone to antibody production. To confirm the ability of AX09 to induce a strong anti-xCT antibody response, we repeated the immunizations using the optimal dose regimen (i.e., 10 µg three times at 2 week intervals) on the Th1-biased strain, C57Bl6 [44]. We also tested the AX09 on the outbred CD-1 mice, to better represent human genetic heterogeneity, obtaining similar results (Figure 2C, black bars).

Since the mice were immunized with VLPs displaying ECD3 from human xCT, we also tested the ability of these sera to react with the mouse ECD3 sequence. Indeed, they were able to bind mouse ECD3 peptide (Figure 2C, white bars), albeit with lower affinity compared to the human ECD3, as shown by the results of the chaotropic ELISA (Figure 2D). Sera from the MS2 wt immunized mice bound neither human nor mouse ECD3. All these data demonstrate the ability of the AX09-induced antibody to bind the linear ECD3 peptide. To determine whether the antibodies were also able to recognize ECD3 in its three-dimensional conformation, we tested the sera from untreated BALB/c mice or BALB/c mice immunized with AX09 or MS2 wt in ELISA using either AX09 or MS2 wt VLPs as coating antigens. As shown in Figure 2E, AX09 immunization induced a high titer of antibody able to specifically bind the ECD3 in its tridimensional conformation adopted in the AB loop on the VLP surface, which is closer to the one naturally expressed on cell membrane. To determine whether the antibodies can potentially recognize ECD3 on xCT in its native environment, we compared the purified IgG from the sera of AX09 and MS2 wt immunized mice for their ability to bind the xCT naturally expressed on the surface of murine tumor cells, using xCT-silenced cells as a control. Spheroids generated either from 4T1 or 4T1 xCT-silenced cells were incubated with purified IgG from the sera of immunized mice. As shown in Figure 2F, while no signal was observed in 4T1 cells incubated with MS2 wt-induced IgG (upper left panel), AX09-induced IgG were able to stain spheroids generated from these xCT-expressing cells (lower left panel). The binding we observed was specific for xCT since we observed positive signals in the 4T1 spheroids silenced with shRNA-control (upper right panel), while no signal was observed in the shRNA-xCT silenced spheroids (lower right panel). Similar results were obtained with spheroids from human MDA-MB-231 tumor cells. To evaluate whether the ability of AX09-induced antibody to bind normal tissue expressing xCT (e.g., cerebellum) can result in toxicity in immunized mice, we looked for signs of neuropathy in the CA3 subfield of the hippocampus with high xCT expression [45]. The CA3 region from AX09-immunized mice was evaluated by immunohistochemistry for the expression of the glial fibrillary acidic protein (GFAP) and the ionized calcium-binding adapter molecule (IBA)-1. As shown in Figure 2G, no neuronal abnormalities were observed. The pyramidal neurons in the CA3 region were arranged neatly and tightly, and no neuron loss was found. Additionally, the neurons in both sections were round and intact with nuclei stained clear, dark blue.

Overall, these data demonstrate that AX09 vaccine administration in mice elicits a strong antibody response against the human ECD3, mirrored by a strong but lower-affinity antibody response to the mouse ECD3, without inducing toxicity.

### 2.3. AX09-Induced Antibody Response Affects Breast Cancer Stem Cell Function In Vitro

To evaluate the ability of AX09-induced antibody to affect xCT function and alter BCSC biology in vitro, purified IgG from the sera of AX09 -and MS2 wt-immunized mice was incubated for 5 days with tumorspheres generated from TUBO, 4T1 and MDA-MB-231 cells; then, functional and morphological parameters were analyzed. The IgG from MS2 wt-treated mice did not influence BCSC self-renewal (Figure 3A) and proliferation capacity (Figure 3B), measured as the number and dimension of tumorspheres, respectively. Instead, the IgG purified from the sera of AX09-immunized mice reduced both the number (Figure 3A) and the dimension (Figure 3B) of the tumorspheres derived from TUBO cells (left panels), exerting an effect comparable to that of the xCT inhibitor sulfasalazine (SAS, positive control). This reduced tumorsphere-generative ability was accompanied by both a decrease in the percentage of BCSCs, evaluated as the percentage of cells positive for aldehyde dehydrogenase-1 activity (Aldefluor^+^ cells; Figure 3C, left panel), and an increase in intracellular ROS (Figure 3D, left panel). This latter parameter is an indication of xCT functional inhibition. Similar results were obtained on tumorspheres derived from 4T1 (Figure 3A–D, central panel) and MDA-MB-231 (Figure 3A–D, right panel) cells.

### 2.4. AX09-Induced Antibody Response Inhibits the Cystine Uptake, Migration and Adhesion of Breast Cancer Cells In Vitro

To further elucidate the potential mechanism of xCT inhibition mediated by the AX09-induced antibody, we evaluated the cystine uptake and the migration ability of tumor cells incubated with IgG from AX09-immunized mice. It is known that xCT exchanges intracellular glutamate with extracellular cystine; therefore, the functional inhibition of xCT should result in a decreased cystine uptake. Spheroids generated from human MDA-MB-231 cells were incubated with purified IgG from immunized mice, then with cystine-FITC, and finally, the amount of cystine uptake was measured. As reported in Figure 4A, purified IgG from mice immunized with AX09 significantly decreased the ability of tumor cells to take up cystine-FITC as compared to that from MS2 wt mice, suggesting that the binding of AX09-induced antibodies to xCT hampers its cystine–glutamate antiporter function. Since xCT-dependent glutamate secretion promotes breast cancer cell migration and invasion [12], we evaluated the effects of antibody-mediated xCT inhibition on tumor cell migration in transwell assays. To this end, MDA-MB-231 (Figure 4B) and 4T1 cells (Figure 4C) were seeded in the upper chamber of the transwell and incubated with sera from untreated mice and mice immunized with MS2 wt and AX09; after 48 h, the number of cells that had migrated to the lower surface of the transwell were counted. As reported in Figure 4B, the AX09-induced antibody strongly reduced the migration ability of the MDA-MB-231 cells as compared to the controls (sera from MS2 wt and untreated mice). Similar results were observed also for 4T1 cells (Figure 4C).

Finally, to further evaluate the effect of AX09-induced antibody on the adhesion/spreading capacities of breast cancer cells, spheroids generated from MDA-MB-231 cells were assessed by a quantitative/dynamic monitoring cell system (xCELLigence Real-Time Cell Analysis, RTCA) in the presence of antibodies from mice immunized with AX09 or MS2 wt. As reported in Figure 4D, purified IgG from the sera of mice immunized with AX09 decreased the ability of tumor cells to spread out from the spheroid to the surrounding area.

### 2.5. AX09-Induced Antibody Response Inhibits Formation of Lung Metastases in Breast Cancer Models

Since the AX09-induced antibody inhibition of xCT function in vitro impairs breast cancer functions including migration, we speculated that AX09 vaccination could result in an inhibition of metastasis formation in vivo. To test this hypothesis, we used two metastatic mammary mouse models. The first one was based on the intravenous (i.v.) injection of TUBO-derived tumorspheres, enriched in BCSCs [9]. Female BALB/c mice were vaccinated with AX09 and MS2 wt (as control) three times at two-week intervals or left untreated. One week after the last boost, the mice were challenged with TUBO cells (Figure 5A). Twenty-one days after tumor challenge, the mice were culled and the lungs removed and weighed; lung weight was assumed as an indication of metastatic burden. As shown in Figure 5B, the lung weights of untreated and MS2 wt-immunized mice were similar, while the lungs of mice treated with AX09 displayed a significant decrease. We then tested the same vaccination protocol (Figure 5A) in a second metastatic model based on BALB/c mice subcutaneously (s.c.) injected with cells derived from 4T1 tumorspheres. While giving rise to a tumor at the site of injection (the primary tumor), the 4T1 cells spontaneously metastasize to the lung, offering the opportunity to evaluate both primary tumor incidence and lung colonization in a way as similar as possible to a clinical setting. Mice immunized with AX09 showed a statistically significant delay in primary tumor onset (Figure 5C) and decrease in lung weight (Figure 5D) as compared to control and untreated mice, confirming the anti-metastatic effect of AX09 vaccination in vivo. In addition to the direct effect of antibodies against xCT, other immune-mediated mechanisms can play a role in the tumor-inhibiting effects observed following AX09 vaccination. To investigate this issue, we characterized, by FACS analysis, the immune infiltrate in the primary tumor and in the lungs of mice left untreated or vaccinated with AX09 or MS2 wt. As shown in Figure 5E, the lungs from mice vaccinated with AX09 displayed a statistically significant increase in the number of T lymphocytes, as compared to untreated mice. This increase in T cells was due to both CD4^+^ and CD8^+^ (Figure 5F). A similar trend was found in the primary tumor (Appendix A). Finally, we also tested the ability of sera from immunized mice to induce ADCC and CDC, but no difference was found compared to controls.

### 2.6. AX09 Vaccination Inhibits Tumor Progression and Metastatic Formation in a Transgenic Mouse Model of Breast Cancer

To evaluate the anti-metastatic and the anti-tumor effect of AX09 in a clinically meaningful setting, transgenic BALB-neuT mice that spontaneously develop mammary tumors and lung metastases were vaccinated with AX09 or MS2 wt six times over a 30 week period of time (Figure 6A). Vaccination started at 7 weeks of age, when the mammary glands already display clearly pre-neoplastic lesions (atypical hyperplasia and in situ carcinomas) due to the over-expression of the rat Her-2 transgene [46]. The antibody response against xCT evaluated two weeks after the second immunization (Figure 6B), was comparable to that of wt BALB/c mice (Figure 2A). The weekly monitoring of mice demonstrated a delayed mammary tumor onset (Figure 6C) and growth (Figure 6D) in AX09-vaccinated BALB-neuT mice as compared to controls. At the end of the experiment, the mice were culled, and their lungs, explanted and weighed. Since no difference in lung weight was observed between the two experimental groups, we proceeded to the enumeration of superficial lung metastases, finding a significant decrease in AX09- as compared to MS2 wt-immunized mice (Figure 6E).

These data demonstrate that AX09’s anti-tumor and anti-metastatic effects are evident also in the very aggressive and stringent BALB-neuT model.

### 2.7. AX09 Vaccination Inhibits Metastasis Formation in a Humanized Breast Cancer Model

To test in vivo AX09’s effect on metastatic human breast cancer, immunodeficient NSG (NOD Scid Gamma) mice were s.c. injected with MDA-MB-231 cells and treated intraperitoneally with sera from mice left untreated or vaccinated with AX09 or MS2 wt, four times, starting 3 days after cell challenge (Figure 7A). Mice were culled 32 days after challenge, tumors and lungs were weighed, and the lung metastases were counted. The mice treated with sera from AX09-immunized mice showed a strong reduction in tumor weight compared with the mice treated with sera from untreated mice, while only a decreased trend was observed in mice treated with sera from AX09-immunized mice compared to mice treated with sera from MS2 wt-immunized mice (Figure 7B). No difference was observed in lung weight between the groups; however, AX09-induced antibodies reduced the number of superficial lung metastasis (Figure 7C) compared to controls, confirming the efficacy of AX09 treatment in inhibiting metastatic progression also in a human breast cancer model, as reported in the representative histological images (Figure 7D). These data support the promise of AX09 for the treatment of MBC.

## 3. Discussion

As extensively reported by several authors and our group, xCT presents the following features as a good target for cancer therapy: (1) it is an oncogenic driver for tumor progression [32]; (2) its increased expression has been correlated with breast cancer invasiveness [12] and epithelial-to-mesenchymal transition [47]; (3) elevated levels of this protein are observed in a high percentage of invasive mammary ductal tumors including triple negative breast cancer (TNBC) and correlate with poor overall survival [1]; and (4) its expression is restricted to only a few normal cell types (such as neurons), and xCT knockout mice show no developmental or growth defects [48].

In previous work [36], we described a first generation VLP vaccine against xCT. Based also on MS2 platform, it displayed the ECD6 of xCT. The ECD6 peptide shows 100% homology between human and mouse sequences and was successfully used against several mouse cancer models. The vaccine impaired tumor progression and the formation of metastases and was well-tolerated. Based on these promising results, we designed a second generation VLP, AX09, which displays the ECD3 of human xCT. We utilized ECD3 mainly for two reasons: first, the sequence is longer than that of ECD6 and should result in the induction of a more diverse oligoclonal antibody response; second, it has been recently demonstrated by Sharma and collaborators [49] that the residues critical for the binding of cystine and glutamate are located in the eight transmembrane domain of xCT, which is consecutive to ECD3. Our hypothesis is that the binding of ECD3 by AX09-induced antibodies blocks xCT’s interaction with cystine and glutamate, thus inhibiting its antiporter function.

Here, we showed that AX09 induces a strong anti-xCT antibody response, displaying a higher affinity for the human than the mouse ECD3. These results were to some extent expected, considering that the human ECD3 has 75% homology with the corresponding murine sequence. Nevertheless, these antibodies reduce self-renewal capacity and alter cellular redox balance in BCSC-enriched tumorspheres from both human and murine breast cancer cell lines. These effects are a consequence of the impairment of xCT function, as shown by the robust reduction of the cystine uptake and reduced migrative ability of breast tumor cells incubated with AX09-induced antibodies. Moreover, vaccination with AX09 reduces metastasis formation and delays tumor onset in different mammary mouse models, including BALB-neuT mice [46]. In addition, AX09-induced antibodies clearly reduce metastasis in an immunodeficient metastatic human breast cancer model.

Based on the data collected in this paper, we have hypothesized the mechanism of action of our therapeutic approach activated by AX09 vaccination (Figure 8). Indeed, we can speculate that cancer cells expressing xCT can disseminate from the primary tumor and migrate to distant organs, giving rise to metastasis formation. AX09 immunization stimulates a strong and specific antibody response that blocks xCT function and reduces metastasis formation.

## 4. Materials and Methods

### 4.1. Production and Stability of VLPs

The sequence of the third domain of human xCT protein was inserted genetically into the AB-loop of MS2 CP as previously described [50] to produce the AX09 VLP. Briefly, the ECD3 sequence was cloned in the pDSP62 plasmid, which was electroporated into the *E. coli*, T7 expression strain C41 (DE3) (Lucigen, Middleton, WI, USA), grown to mid-log phase. VLP expression was induced by the addition of isopropyl β-d-1-thiogalactopyranoside (IPTG, 1 mM, Sigma-Aldrich, St. Louis, MO, USA) for 3 h at 28 °C. Bacterial pellets were lysed in lysis buffer (50 mM Tris-HCl, pH 8.5, 100 mM NaCl, 10 mM EDTA, Sigma-Aldrich), sonicated and purified from bacterial debris by centrifugation. Bacterial DNA was removed by treating the supernatant with DNase I (10 units/mL, 1 h at 37 °C; Sigma-Aldrich), and the VLPs were purified by size exclusion chromatography (Sepharose CL-4B resin, Sigma-Aldrich). The buffer exchange (Phosphate-buffered saline, PBS) and concentration of purified VLPs were achieved using ultrafiltration (Amicon Ultrafiltration device, 100 Kd MWCO, Sigma-Aldrich), and the resulting VLP preparation was quantified by the Bradford assay (BioRad, Philadelphia, PA, USA). VLP purity was assessed by agarose and SDS-PAGE gel electrophoresis. By creating the single-chain dimer technology, the thermodynamic of the generated VLP is increased, making the VLP more tolerant to peptide insertion and more resistant to degradation. The subunit fusion allows the production of a VLP stable up to at least 50 °C. To compare the thermal stability, we determined a denaturation profile by measuring the fraction of the protein remaining soluble after 2 min at a given temperature. The VLP was suspended at a concentration of 1 mg/mL in 10 mM tris-HCl, and 0.1 mM MgCl_2_ (Sigma-Aldrich), pH 7.2 was added to pre-heated tubes at a series of specified temperatures. After 2 min, the tubes were transferred to ice, where they remained for a few minutes, and then centrifuged for 5 min at ~13,000 rpm. The Bradford assay was used to measure the protein amount in both the insoluble and soluble fractions. The relative rates of the protein denaturation at a fixed temperature of 55 °C were determined by a similar method. AX09 was also tested from one to ten cycles of freezing and thawing. The protein was frozen at −20 °C and then thawed at room temperature for two hours, and the tubes were returned to the freezer according to the number of freeze and thaw cycles. By the end of the experiment, the soluble part containing the VLP was analyzed by electrophoresis in a 1% agarose gel with ethidium bromide.

### 4.2. Cells, Spheroids and Tumorspheres—Generation

MDA-MB-231 cells were grown in Dulbecco’s Modified Eagle Medium (DMEM, Invitrogen, Monza, Italy) supplemented with 10% heat-inactivated fetal bovine serum (FBS, Sigma-Aldrich). 4T1 cells were grown in RPMI-1640 medium (Invitrogen) supplemented with 10% FBS. All the cells were purchased from ATCC (LGC Standards, Sesto San Giovanni, Italy). TUBO cells were cultured in DMEM supplemented with 20% FBS. Each culture medium was supplemented with penicillin (100 U/mL) and streptomycin (100 U/mL) (Sigma-Aldrich). The cells were cultured at 37 °C in a 5% CO_2_ incubator and tested negative for mycoplasma using the polymerase chain reaction (PCR) MycoAlert assay (Lonza, Basel, Switzerland). Tumorspheres were generated as previously described [51]. For the spheroid generation, cells were detached with trypsin 1× (Sigma-Aldrich), washed and suspended at 20,000 cells/mL in DMEM-F12 (Invitrogen) supplemented with 10% FBS. Then, the cell suspension was mixed 1:1 in complete medium with 0.4% methylcellulose (Sigma-Aldrich) and plated at 100 µL/well in ultra-non-adherent 96-well suspension culture plates (CellStar, Greiner Bio-One, Frickenhausen, Germany).

### 4.3. Purification of IgG for In Vitro Assays

Female BALB/c mice were administered 10 μg of VLP by intramuscular (i.m.) injection followed by two boosts 2 weeks apart. One week after the final boost, blood was collected by cardiocentesis and immune sera were isolated as described [52]. Total IgG was purified from the sera using affinity chromatography (Protein A/G Plus Agarose, Thermo Fisher Scientific, Carlsbad, CA, USA) following the manufacturer’s protocol. Eluted fractions containing IgG as assessed by SDS-PAGE analysis were concentrated into PBS using ultrafiltration (Amicon ultrafiltration device, 10 Kd MWCO), and the protein concentration was determined by the Bradford assay using a mouse IgG standard curve (BioRad). Purified IgG was aliquoted and stored at −20 °C.

### 4.4. ELISA

Female BALB/c mice were vaccinated i.m. three times at two-week intervals or two times at three-week intervals with different doses of AX09 (2.5, 5, 10 and 20 µg). In another experiment, female BALB/c, C57Bl6 and CD-1 mice were vaccinated i.m. with 10 µg of AX09 or MS2 wt (empty vector, as control) followed by two boosts at a 2 week interval. One week after the final boost, blood was collected, and sera were isolated. Antibody responses were evaluated by ELISA on plates coated with 500 ng of human ECD3 biotinylated peptides absorbed on the wells of NeutrAvidin coated ELISA plates with the blocker BSA (Thermo Fisher Scientific). ELISA was also performed on a plate coated with mouse or human ECD3 xCT peptides (GenScript, Piscataway, NJ, USA, 1 μg/well) or with AX09 or MS2 wt VLP (500 ng/well). After blocking with blocking buffer (1× PBS, 1% BSA + 0.05% Tween-20, Sigma-Aldrich) for 1 h at 37 °C, mouse sera (diluted in blocking buffer) were incubated for 2 h at 37 °C. After 3 washes, HRP-conjugated anti-mouse Total IgG (1:2000, Sigma-Aldrich), IgG2a, IgG2b, IgG1 and IgG3 (1:2000, Life Technology, Monza, Italy) were incubated for 2 h at 37 °C. The reaction was developed with TMB Solution (3′,3′,5′,5′-tetramethylbenzidine, Sigma-Aldrich), then stopped using 2N HCl. Serum reactivity was determined by measuring the optical density (O.D.) at 450 nm using a 680XR microplate reader (BioRad). For the chaotropic ELISA, after the incubation with the sera and before the incubation with secondary antibody, plates were washed and incubated with 0, 2 or 4 M NH_4_SCN (Sigma-Aldrich) chaotropic agent solution for 15 min at room temperature.

### 4.5. Immunofluorescence Analysis

Spheroids derived from 4T1 cells were fixed with 4% paraformaldehyde for 5 min at room temperature, washed with PBS, blocked with 5% BSA for 30 min and incubated overnight at 4 °C with 1 μg/mL of purified IgG from mice immunized with AX09 or MS2 wt. Specific antibody binding was detected using Alexa flour 488-conjugated goat anti-mouse Ig (1:400, Thermo Fisher Scientific), and nuclei were stained with DAPI (1:10000, Thermo Fisher Scientific). Images were acquired with a Leica Fluorescence Microscope (Leica, Buffalo Grove, IL, USA) with 40× magnification. Immunofluorescence on silenced 4T1 was performed using lentiviral vectors (pCCLsin.PTT.PGK.EGFP.Wpre, pMDLg/pRRE, pRSV-Rev and pMD2.VSVG), as described [53]. The following sequences were used: CGGAAATCCTCTCTATGATTC for mouse and CTGATTTATCTTCGATACAAC for human xCT. Briefly, HEK293FT cells were transiently transfected (Lipofectamine 2000; Invitrogen) for 16 h, then lentiviruses were harvested 24 and 48 h later and filtered through 0.22 µm-pore cellulose acetate filters. Recombinant lentiviruses were concentrated with Amicon Ultra-15 Centrifugal Filter Units (NMWL of 30 kDa). Lentiviral vector viability was confirmed by reporter gene expression and drug selection.

### 4.6. Immunized Sera’s Effects on Tumorspheres

Tumorspheres were dissociated and cultured at a density of 6 × 10^4^ cells/mL in a 6-well dish in the absence or presence of 50 µg/mL of purified IgG or of 50 µM SAS (sulfasalazine; Sigma-Aldrich). After 5 days, the tumorspheres were imaged with an ApoTome fluorescence microscope (Zeiss, Oberkochen, Germany) and sphere diameter measured with the AxioVision 4.8 software as described [36]. Spheres were counted and reported as the number of spheres generated for every 10^3^ cells plated, then they were dissociated and processed for the FACS analysis of Aldefluor and ROS.

### 4.7. Cell Migration Assay

To measure cell migrative ability [54], 2 × 10^4^ MDA-MB-231 and 4T1 cells were seeded in the top chamber of a 24-transwell plate (8 μm pore size; Corning, Amsterdam, The Netherlands) after 1 h of pre-incubation with 1:50 dilutions of pooled sera from mice vaccinated with AX09 or MS2 wt or left untreated in a serum-free medium. Then, the bottom chambers of the transwell plates were filled with 600 μL of complete growth medium and the cells were incubated at 37 °C in a 5% CO_2_ atmosphere. After 48 h, the non-migrated cells on the top side of the filter were removed by scrubbing twice with a cotton-tipped swab. Migrated cells on the bottom side of the filter were fixed with 2.5% glutaraldehyde (Sigma-Aldrich) and stained with 0.2% crystal violet (Sigma-Aldrich). After washing, the migrated cells of four randomly selected fields per well were imaged using an Olympus BX41 microscope (Olympus Corp., Tokyo, Japan) and analyzed using the Fiji and ImageJ Software (Rasband, W.S., ImageJ, US National Institutes of Health, Bethesda, MD, USA).

### 4.8. Cystine Assay

MDA-MB-231 cells were starved in 1% BSA/DMEM (without methionine, cystine or cysteine) and pretreated with purified IgG from the sera of BALB/c mice vaccinated with AX09 or MS2 wt for 60 min, followed by incubation with 1μM cystine-FITC (Millipore, Darmstadt, Germany) in 5% FBS-dialyzed/DMEM for 8 h. The cells were fixed and stained with DAPI (Thermo Fisher Scientific). Cystine uptake was evaluated by subjecting samples to fluorescence microscopy for the quantification of the mean intensity of cysteine-FITC fluorescence in >1000 cells in images acquired with the Cytation 5 cell imaging and analysis software (Gen5 by counting cell-FITC-green/total cells DAPI).

### 4.9. Spheroid Spreading/Adhesion Assay

Spheroids were pretreated with sera from vaccinated BALB/c mice for 60 min and directly transferred to E-Plate (ACEA Biosciences, San Diego, CA, USA) in the presence of RPMI with 5% of FBS and monitored over time with an xCELLigence Real-Time Cell Analysis (RTCA) system for adhesion/spreading (for 24 h). Cell adhesion and spreading (reported as cell index) were monitored over time (one read every 15 min) with an xCELLigence Real-Time Cell Analysis (RTCA) system (ACEA Biosciences). Fluorescence microscopy images of spheroids treated with antibodies from mice immunized with AX09 or control MS2 wt were captured after 24 h of growing and staining with Phalloidin (Sigma-Aldrich) and DAPI (1:10,000, Thermo Fisher Scientific).

### 4.10. Histology of the Brain

CD-1 mice were i.m. vaccinated three times at two-week intervals, and at 11 days after last boost, they were culled. The brains were harvested and fixed in 10% formalin (Sigma-Aldrich). After 36 h, the brains were sliced transversely into 7 sections, 1–2 μm thick, and processed for the histology analysis. Fixed sections were incubated at room temperature for 1 h in 5% normal goat serum in PBS, followed by overnight incubation at 4 °C with Iba-1 antibody (1:100; Abcam, Burlingame, CA, USA) for microglia and GFAP antibody (1:300; Cell Signaling, Danvers, MA, USA) for astrocytes. Thereafter, the sections were washed three times with PBS and incubated for 1 h with goat anti-rabbit IgG (1:500, Abcam) and goat anti-mouse IgG (1:500, Abcam). Positive cells were quantified under a laser scanning confocal microscope in stained sections in the hippocampus and cortex.

### 4.11. In Vivo Treatments

Female BALB/c mice (Charles River Laboratories, Sant’Angelo Lodigiano, Italy) were vaccinated i.m. three times at 2 week intervals before tumor challenge (preventive model). Each mouse received 10 μg of AX09 or MS2 wt (as control) formulated in PBS for a total volume of 50 μL. In a set of experiments, mice were i.v. challenged one week after last vaccination with 5 × 10^4^ tumorspheres derived from TUBO cells and sacrificed 21 days later. In the other set of experiments, mice were injected s.c. one week after last vaccination with 1 × 10^4^ P1 4T1 tumorsphere-derived cells in the left flank. Tumors were measured twice a week with calipers by two perpendicular diameters. Tumor growth was reported as tumor mean diameter. When the tumor reached a mean diameter of 10 mm, mice were euthanized. At sacrifice, lungs were removed and weighed. BALB-neuT [55] mice were bred under specific pathogen-free conditions (Allentown Caging Equipment, Allentown, NJ, USA) at the Molecular Biotechnology Center of the University of Torino, Italy. Female BALB-neuT mice were immunized with 10 μg of AX09 or MS2 wt six times. The first two immunizations were performed two weeks apart starting from Week 7 of age, and then mice were vaccinated monthly four times. Sera were collected two weeks after the second immunization. To evaluate mammary tumor incidence, BALB-neuT females were inspected weekly by palpation, and progressively growing masses with a mean diameter of >1 mm were considered as tumors. Each week, mice were inspected, and tumor mass was measured with a caliper. Growth was monitored until all 10 mammary glands displayed a tumor or until a tumor exceeded a mean diameter of 10 mm, at which point mice were sacrificed for humane reasons. At the end of the experiment, lungs were explanted and weighed and the superficial lung metastases were counted using a stereomicroscope (Semi DV4 SPOT, Zeiss). Immunodeficient NSG female mice (Charles River Laboratories) were challenged subcutaneously with 1 × 10^6^ MDA-MB-231 cells in the left flank and then treated intraperitoneally (i.p.) 4 times (3, 6, 10 and 14 days after cell injection) with 200 µL of pooled sera from mice vaccinated with AX09 or MS2 wt and from untreated mice. Tumor growth was measured with a caliper twice a week. The mice were culled when the subcutaneous tumors of control mice reached 10 mm in mean diameter. At the end of the experiment, subcutaneous tumors and lungs were explanted and weighted. Superficial lung metastases were counted; lungs were harvested and fixed in 10% neutral buffered formalin (Bio-Optica, Milan, Italy), paraffin-embedded, sectioned, and stained with hematoxylin and eosin (Bio-Optica) for metastasis detection; and images were acquired with a Leica DMRD optical microscope (X25 microscopic fields).

### 4.12. FACS Analysis

Tumorsphere-derived cells were stained with the Aldefluor kit (Stem Cell Technologies, Grenoble, France) as reported [51]. To measure intracellular ROS content, cells were stained with 2′, 7′-dihydrochlorofluorescein diacetate (DHCF-DA, Sigma-Aldrich) as described [9]. The results were expressed as percentages of positive cells and as mean fluorescence intensity (MFI). For the analysis of immune infiltrates, lungs and tumors from vaccinated mice were finely minced with scissors and then digested by incubation with 1 mg/mL of collagenase IV (Sigma Aldrich) in RPMI-1640 (Life Technologies) at 37 °C for 1 h in an orbital shaker. After washing in PBS, cell suspensions were incubated with a buffer for erythrocyte lysis (155 mM NH_4_Cl, 15.8 mM Na_2_CO_3_, 1 mM EDTA, pH 7.3) for 10 min at room temperature, then passed through a 40 µm-pore cell strainer, centrifuged at 1800 rpm for 10 min and suspended in PBS. The cells were treated with an Fc receptor (FcR) blocker (CD16/CD32; Becton Dickinson) for 5 min at 4 °C and then stained for 30 min at 4 °C with the following Abs: anti-mouse CD45 VioGreen, anti-mouse CD3 FITC, anti-mouse CD4 APC-Vio770, anti-mouse CD8 VioBlue, and anti-mouse CD49b PE (all from Miltenyi Biotec, Bologna, Italy). All samples were acquired on a BD FACSVerse and analyzed with the BD FACSuite™ software (Becton Dickinson, Milan, Italy).

### 4.13. Statistics

Differences in sphere formation, diameter, antibody binding, lung weight, metastasis number, and the number of infiltrating cells were evaluated by Student’s *t*-test. The Mantel–Cox test was used for the statistical analysis of tumor incidence (GraphPad Software, San Diego, CA, USA). Data are reported as mean ± SEM unless otherwise stated. Values of *p* ≤ 0.05 were considered significant.

### 4.14. Study Approval

For the experiments performed at the Molecular Biotechnology Center, mice were treated according to the European Guidelines and policies, as approved by the Italian Ministry of Health and the University of Torino Ethical Committee (CC 652.28; DGSAF 22132-A; CC 652.45). The experiments conducted at University of Texas at El Paso were performed according to the IACUC Guidelines (protocol number A-201201-1). Some experiments were performed in Nanospectra (Huston, TX, USA) according to the IACUC Guidelines (protocol number 2017-001).

## 5. Conclusions

Overall, our data reinforce the promise of xCT immune targeting as a therapy for MBC and support the development of AX09 for potential clinical use as a second-line (or later) treatment as an adjunctive therapy in combination with chemotherapy or with PD-1/PD-L1 checkpoint inhibitors. In particular, association with chemotherapy could be a promising treatment since cancer cells upregulate xCT as a mechanism of self-defense in response to many chemotherapeutic drugs, and xCT inhibition sensitizes tumor cells to chemotherapy in vitro [56,57]. Moreover, it has been demonstrated in pre-clinical mouse models of breast cancer that the combination of a DNA vaccine against xCT with doxorubicin increases the anti-metastatic and anti-tumor effects of the single treatments [9]. Finally, the potential association of AX09 with chemotherapy supports the high translatability of our vaccine to human patients, since chemotherapy is one of the leading treatments for women affected by MBC.

To this end, we have already evaluated the feasibility of a scale-up production of AX09 and demonstrated that AX09 has a high thermostability. AX09 storage does not require a stabilizing protein cocktail, cryoprotectants, protease inhibitors or anti-microbial agents. This guarantees high vaccine purity and easy storage conditions, which are important features for the translation to human patients. Thus, AX09 may offer several advantages over other therapeutic approaches as a potentially powerful combination therapy to achieve durable responses for MBC patients.

## 6. Patents

AX09 VLP is covered by US patent number 10,588,953 issued on 17 March 2020.

## Figures and Tables

**Figure 1 cancers-12-01492-f001:**
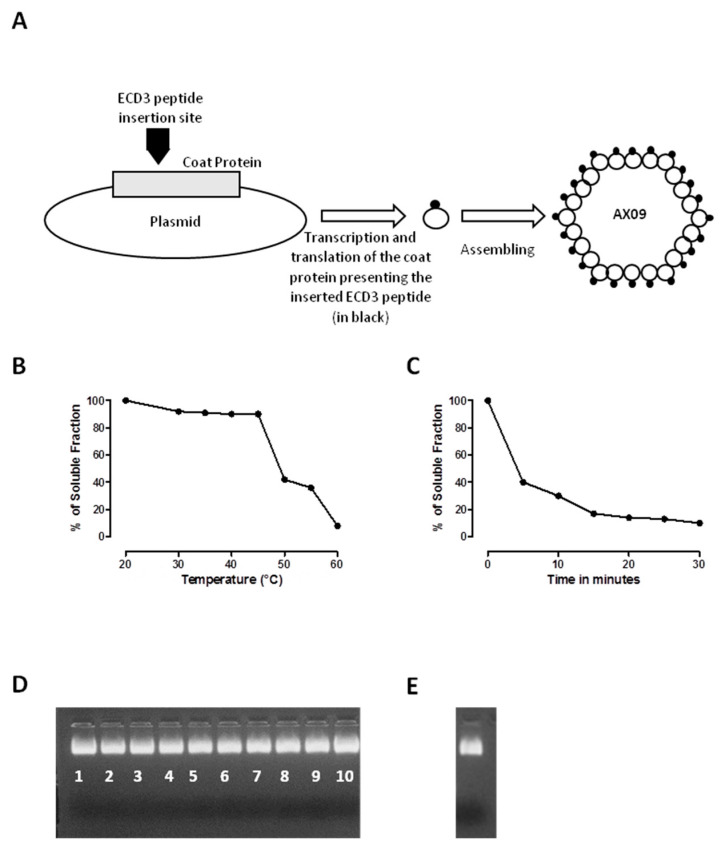
Virus-like particle (VLP) production and manufacturing. (**A**) The basic features of the plasmid used in this study for the synthesis of the AX09 VLP. The sequence of the third domain of the human xCT protein was inserted in the AB-loop of the MS2 coat protein using a pDSP62 plasmid. The resulting modified protein was produced by expression in *E. coli* and purified by Sepharose CL-4B column chromatography. (**B**–**E**) Evaluation of AX09’s thermostability. Percentage of the soluble fraction after (**B**) denaturation at increasing temperature or (**C**) denaturation as a function of the time at 55 °C. The values shown are the averages of two independent measurements. Images of 1% agarose gel electrophoresis of the soluble fraction after exposure to repeated cycles of freezing (−20 °C) and thawing (room temperature), with the number of freeze–thaw cycles reported under the bands (**D**), or storage for 4 weeks at 4 °C (**E**). The bands represent the intact AX09 stained with EtBr.

**Figure 2 cancers-12-01492-f002:**
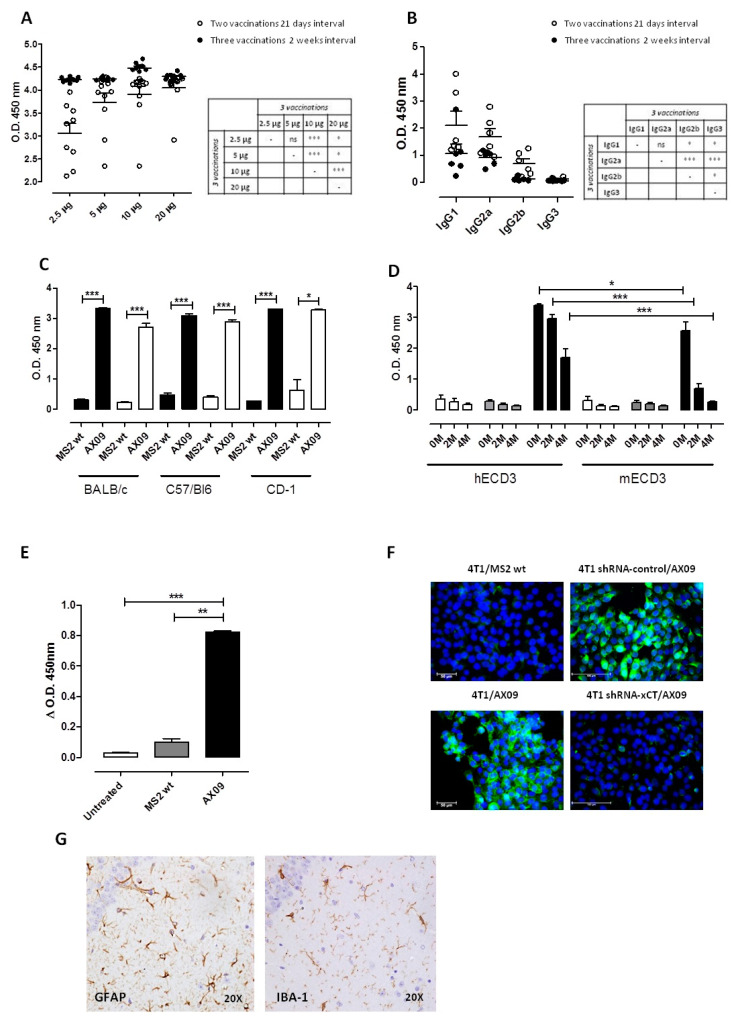
Evaluation of the antibody response against xCT induced by AX09 immunization. Optimization of the VLP doses and protocol. (**A**) BALB/c mice were vaccinated twice at a 21 day interval (white dots) or three times 2 weeks apart (black dots) with different doses of AX09. An ELISA assay was performed using 1:1500 serum dilutions in plates coated with biotinylated human ECD3 peptide, and the signal was detected using horseradish peroxidase-labeled goat anti-mouse IgG secondary antibody followed by development with TMB. Reactivity was measured by the optical density (O.D.) at 450 nm. (**B**) Characterization of the IgG subclasses in mice immunized with 10 μg of AX09 two or three times as described above. An ELISA assay was performed using 1:1000 serum dilutions. In the graphs (**A**,**B**), each dot represents a single mouse. (**C**) ELISA assay coating the plate with human (black bar) or mouse (withe bar) ECD3 peptide using 1:50 serum dilutions of MS2 wt- or AX09-immunized mice from different strains (BALB/c, C57Bl6 and CD-1 mice). (**D**) Antibody affinity was assessed by the ELISA assay, testing the sera of AX09- (black bar) or MS2 wt (grey bar)-vaccinated and untreated (white bar) BALB/c mice at 1:50 dilutions. The plate was coated with human (left columns) or mouse (right columns) ECD3 peptide and was incubated with the NH_4_SCN chaotropic agent solution at different concentrations (0, 2 and 4 M) after serum incubation. (**E**) An ELISA assay was performed, coating the plate with AX09 or MS2 wt VLP and incubating with 1:1000 serum dilutions of AX09 (black bar)- or MS2 wt (grey bar)-vaccinated or untreated (white bar) BALB/c mice. The data reported in the graph are obtained by subtracting the O.D. value at 450 nm measured in the plate coated with MS2 wt VLP from those measured in the plate coated with AX09. (**F**) An immunofluorescence assay was performed on spheroids derived from 4T1 (left panels) or from 4T1 silenced with shRNA-CTRL (upper right panels) or shRNA-xC (lower right panels). Cells were incubated overnight at 4 °C with 1 µg/mL of purified IgG from the sera of AX09 (lower left and right panels) or MS2 wt (upper left panel) immunized mice. The xCT binding (green) was detected with Alexa flour 488 conjugated secondary antibody (dilution 1:400 from ThermoFisher), and the nuclei were stained with DAPI (4′,6-diamidino-2-phenylindole, blue). Images were obtained with fluorescence Microscopes, Leica, with 40× magnification. (**G**) Histological analysis of the brain from AX09-vaccinated CD-1 mice. Immunohistochemistry IHC staining for GFAP and IBA-1 were used to examine the neuropathology in the CA3 subfield of the hippocampus (objective magnification 20×). (**A**–**E**) graphs show the mean ± SEM of the O.D. measurement at 450 nm from at least two independent experiments. * *p* < 0.05; ** *p* < 0.001, *** *p* < 0.0001; Student’s *t*-test.

**Figure 3 cancers-12-01492-f003:**
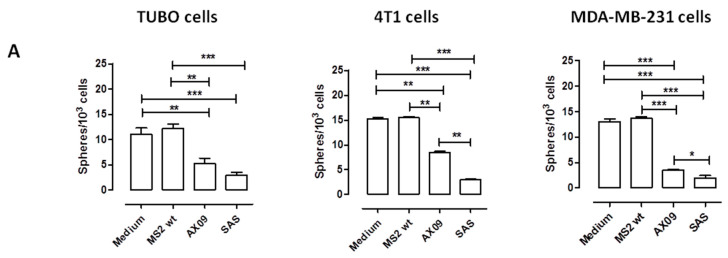
Evaluation of AX09-induced antibody’s effect on breast cancer stem cell (BCSC) biology. Tumorspheres derived from TUBO (left panel), 4T1 (middle panel) and MDA-MB-231 (right panel) cells were incubated for 5 days with medium, purified IgG (50 µg/mL) from the sera of BALB/c mice immunized with MS2 wt or AX09 and the xCT inhibitor sulfasalazine (SAS, 500 µmol). (**A**) Sphere-generating ability described as tumorsphere number/10^3^ seeded cells. (**B**) Sphere diameter was calculated with the AxioVision 4.8 software. FACS analysis of (**C**) Aldefluor positivity reported, as the percentage of positive cells, and of (**D**) ROS production, reported as DCF Mean of Fluorescence Intensity (MFI). Graphs show the mean ± SEM of the values from at least two independent experiments. * *p* < 0.05, ** *p* < 0.001, *** *p* < 0.0001; Student’s *t*-test.

**Figure 4 cancers-12-01492-f004:**
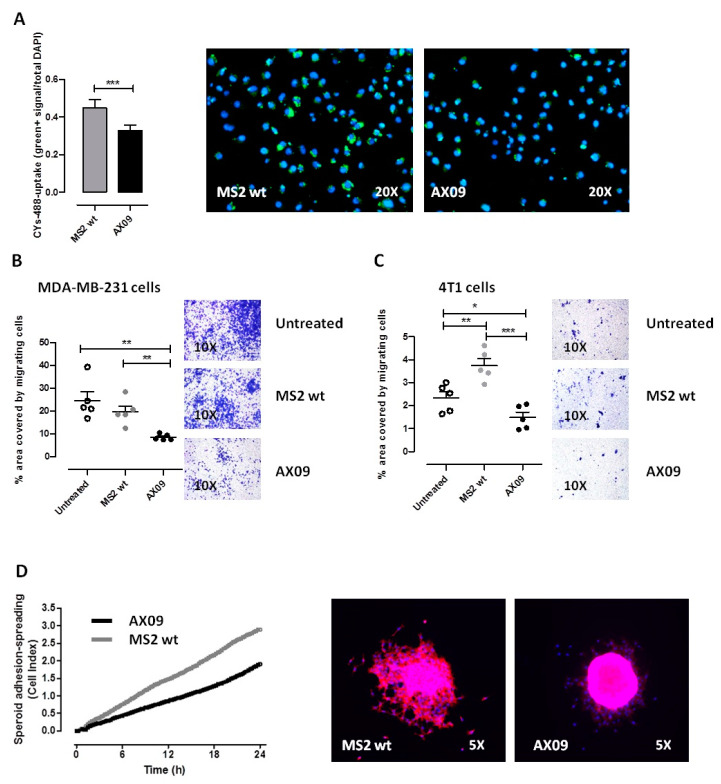
Evaluation of the ability of AX09-induced antibodies to inhibit xCT. (**A**) Cystine-FITC uptake. MDA-MB-231 cells were starved in 1% BSA (Bovine Serum Albumin)/DMEM (Dulbecco’s Modified Eagle’s Medium, without methionine, cystine or cysteine) and pretreated with purified IgG from the sera of mice immunized with AX09 (black bar, right image) or MS2 wt (grey bar, left image) for 60 min followed by incubation with 1 μM of cystine-FITC in 5% fetal bovine serum (FBS)-dialyzed/DMEM for 8 h. Cells were fixed and stained with DAPI. Cystine uptake was evaluated by subjecting the samples to fluorescence microscopy (objective magnification 20X) for the quantification of the mean intensity of cystine-FITC fluorescence in >1000 cells in images acquired with the Cytation 5 cell imaging and analysis software (Gen5 by counting cell-FITC-green/total cells DAPI^+^). (**B**,**C**) Transwell migration assay. MDA-MB-231 (**B**) or 4T1 (**C**) cells were incubated for 60 min at 37 °C under 5% CO_2_ with 1:50 dilutions of sera from BALB/c mice immunized with AX09 (black dots) or MS2 wt (grey dots) or left untreated (white dots). Then, cells were placed in the upper chamber and incubated for 48 h. Cells that had migrated to the lower surface of the membrane were fixed with glutaraldehyde and stained with crystal violet for microscopical observation. Images were taken with the Olympus BX41 microscope (Olympus, objective magnification 10×) and analyzed using the Fiji and ImageJ Softwares. The mean ± SD of the numbers of migrated cells counted in five different fields are reported in the graphs. (**D**) Spheroid spreading/adhesion. Spheroids were pretreated with purified IgG (1 μg/mL) from the sera of immunized mice for 60 min and directly transferred to E-Plate (ACEA Biosciences) in the presence of RPMI containing 5% of FBS and monitored over time with an xCELLigence Real-Time Cell Analysis (RTCA) system for adhesion/spreading (for 24 h). Cell adhesion and spreading (reported as cell index) were monitored over time (one read every 15 min) with an xCELLigence Real-Time Cell Analysis (RTCA) system. Representative images (from fluorescence microscopy, objective magnification 5× of spheroids treated with antibodies from mice immunized with AX09 (black line) or control MS2 wt (grey line) are reported after 24 h of growing and staining with Phalloidin (in red) and DAPI (blue). Graphs show the mean ± SEM of the results from at least two independent experiments. * *p* < 0.05, ** *p* < 0.001, *** *p* < 0.0001; Student’s *t*-test.

**Figure 5 cancers-12-01492-f005:**
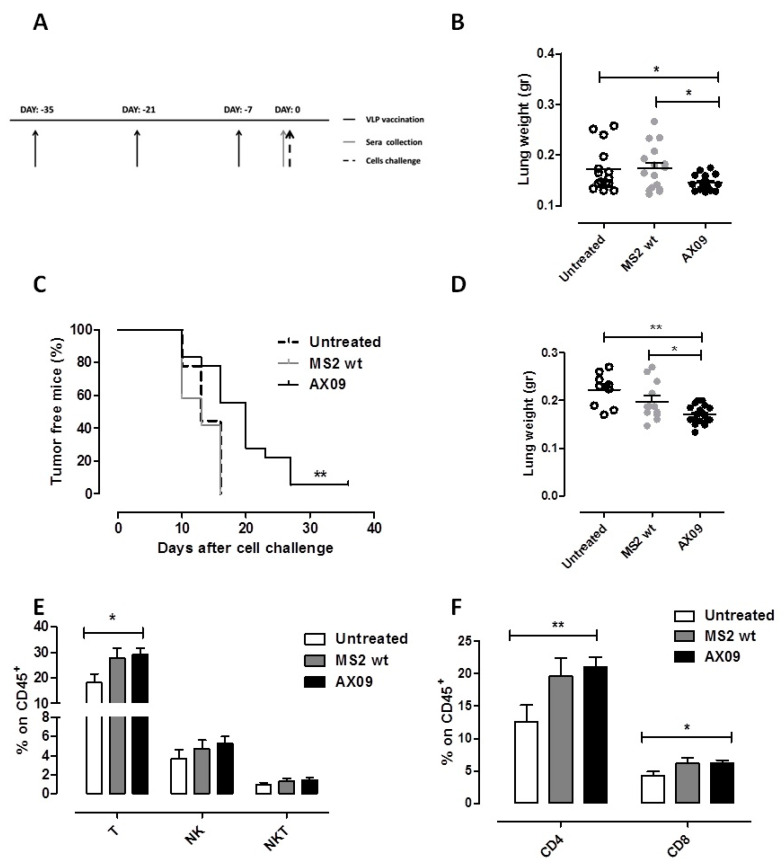
In vivo evaluation of AX09’s anti-metastatic effect in a preventive setting. (**A**) Schematic representation of the preventive protocol. BALB/c mice were vaccinated three times at two-week intervals with AX09 (black) or MS2 wt (grey) or left untreated (white dots or dotted line). One week later, mice were challenged with tumorspheres. (**B**) Metastatic artificial model. Mice (13 untreated, 14 immunized with MS2 wt and 14, with AX09) were intravenously (i.v.) injected with tumorspheres derived from TUBO cells. Twenty-one days after tumor challenge, mice were culled, and the lungs removed and weighed. (**C**–**F**) Spontaneous metastatic model. Mice (9 untreated, 18 immunized with AX09 and 12, with MS2 wt) were subcutaneously (s.c.) injected with tumorspheres derived from 4T1 cells. Tumor incidence was monitored weekly and reported in the Kaplan–Meier graph (**C**) as the percentage of tumor-free mice. When the s.c. tumor reached 10 mm in mean diameter, mice were culled and the lungs explanted, weighed (**D**) and processed for immune infiltrate analysis by FACS. (**E**) Percentage ± SEM of CD45^+^ cells expressing the markers of T (CD3^+^CD49^−^), NK (CD3^−^CD49^+^) and NKT (CD3^+^CD49^+^) cells. (**F**) Percentage ± SEM of CD4^+^ or CD8^+^ cells gated on CD45^+^ cells. All the data are derived from three independent experiments. * *p* < 0.05,** *p* < 0.01; Student’s *t*-test and Mantel–Cox test for the tumor incidence.

**Figure 6 cancers-12-01492-f006:**
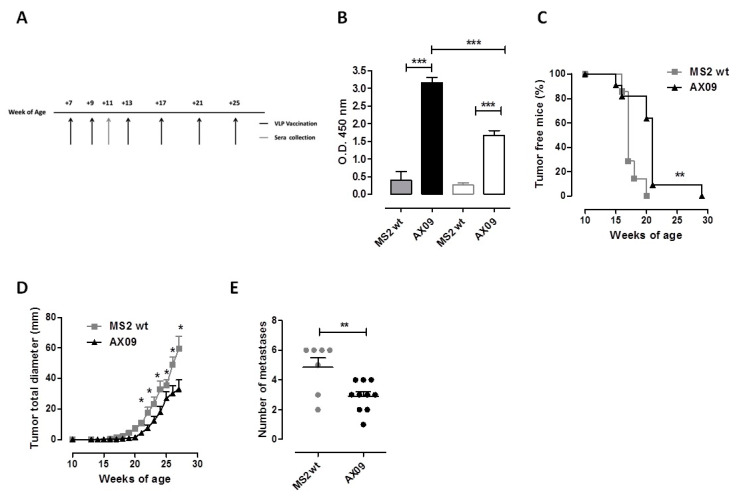
Evaluation of AX09’s anti-metastatic effects in a therapeutic setting. (**A**) Schematic representation of the therapeutic protocol used for BALB-neuT mice treatment. Mice were immunized with AX09 (black bar/line/dots) or MS2 wt (grey bar/line/dots) six times. The first two immunizations were performed two weeks apart, starting from Week 7 of age, and then the mice were vaccinated monthly four times. (**B**) Sera collected two weeks after the second immunization were tested by ELISA for the presence of antibody against human (full bars) and mouse (empty bars) ECD3 peptide. The mice were monitored weekly, and tumor incidence is reported in the Kaplan–Meier graph (**C**) as the percentage of tumor-free mice; tumor growth was measured by a caliper, and the total tumor diameter, calculated as the sum of the single tumor diameter, is shown in the graph (**D**). Mice were culled when at least one mammary tumor reached 10 mm in mean diameter; lungs were removed, and superficial lung metastases were counted (**E**). Graphs show the mean ± SEM of the values; * *p* < 0.05, ** *p* < 0.01, *** *p* < 0.0001; Student’s *t*-test and Mantel–Cox test (for tumor incidence).

**Figure 7 cancers-12-01492-f007:**
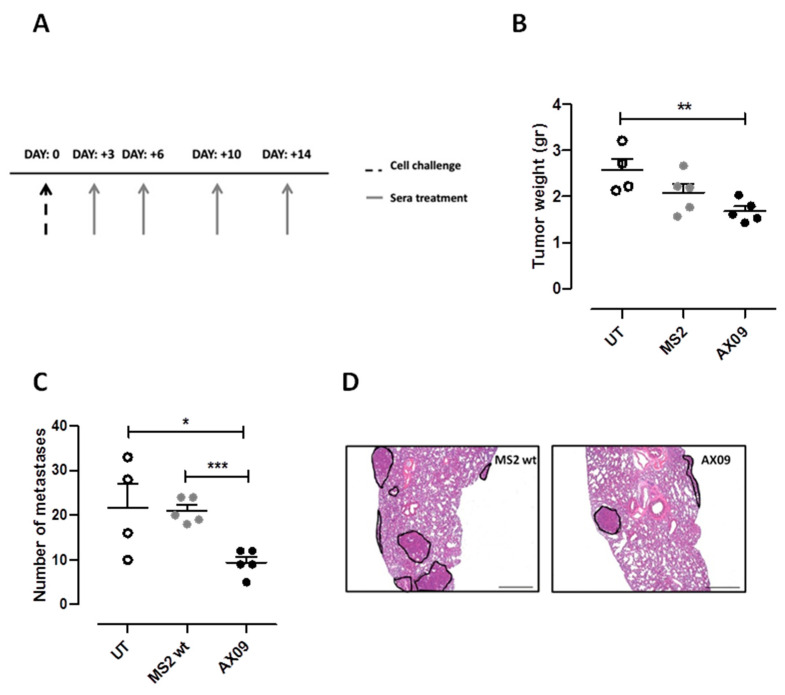
Evaluation of AX09’s anti-metastatic effects in a human breast cancer mouse model. (**A**) Schematic representation of the therapeutic protocol used for immunodeficient NSG mice injected s.c. with 1 × 10 ^6^ MDA-MB-231 cells. Mice were treated intraperitoneally, four times, 3–4 days apart, with 200 µL of pooled sera from immunocompetent BALB/c mice immunized with AX09 (5 mice, black dots) or MS2 wt (5 mice, grey dots) or left untreated (4 mice, with dots). At the end of the experiment, primary tumors were weighed (**B**) and superficial lung metastases were counted (**C**). (**D**) Histological analysis of lung processed for hematoxylin and eosin staining for metastasis detection. The images were acquired with a Leica DMRD optical microscope (×25 microscopic fields). Graphs show the mean ± SEM of the values; * *p* < 0.05, ** *p* < 0.01, *** *p* < 0.0001; Student’s *t*-test.

**Figure 8 cancers-12-01492-f008:**
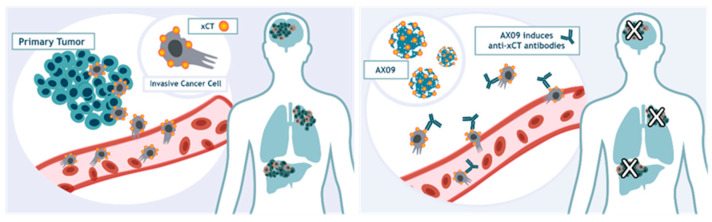
Putative mechanisms of AX09-induced antibodies’ effects on metastasis progression. Schematic representation of the mechanism of action of the AX09-induced antibodies on metastatic progression.

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
