# Peer review of "Development of a VLP-Based Vaccine Displaying an xCT Extracellular Domain for the Treatment of Metastatic Breast Cancer"

_cancers, 2020, doi:10.3390/cancers12061492_

Round 1

Reviewer 1 Report

The manuscript is well organized and easily readable. It deals a very important topic looking for complementary answers to current treatment of metastatic breast cancer.

In my opinion the study is close to acceptable form for publication, I have only some suggestions:

  1. In the introduction it might be useful to better explain the role of xCT in cancer (currently this part is exposed in details at the beginning of discussion)
  2. It can be clearer if a legend is reported directly in the figures (near graphs)
  3. In figure 2G: Is the writing "Dentate Fascia" part of the figure?
  4. In the conclusion  the authors highlighted that their data support the development of AX09 for potential clinical use as a second line (or later) treatment as an adjunctive therapy in combination with chemotherapy or with PD-1/PD-L1 checkpoint inhibitors. Actually in the manuscript the effect of chemotherapy on xCT expression was not reported. Does chemotherapy modulate xCT expression levels?  This point is very important, please discuss it.

Author Response

Response to Reviewer#1 comments

  1. In the introduction it might be useful to better explain the role of xCT in cancer (currently this part is exposed in details at the beginning of discussion)

RESPONSE 1:

We thank the Reviewer for this suggestion. We have added a brief description of the role exerted by xCT in cancer cells in the Introduction, Lines n° 53-59; 65-67.

  1. It can be clearer if a legend is reported directly in the figures (near graphs)

RESPONSE 2:

As suggested, we added figure legends in Fig. 2A and 2B, 4D, 5C, 5E, 5F, 6C, 6D.

  1. In figure 2G: Is the writing "Dentate Fascia" part of the figure?

RESPONSE 3:

To make the figure clearer, we removed all the words from the Fig. 2G, leaving only GFAP and IBA-1, since the explications are reported in the result paragraph and in the figure legend.

  1. In the conclusion  the authors highlighted that their data support the development of AX09 for potential clinical use as a second line (or later) treatment as an adjunctive therapy in combination with chemotherapy or with PD-1/PD-L1 checkpoint inhibitors. Actually in the manuscript the effect of chemotherapy on xCT expression was not reported. Does chemotherapy modulate xCT expression levels?  This point is very important, please discuss it.

RESPONSE 4:

As suggested by the reviewer, we have discussed the effect of chemotherapy on xCT expression and the data obtained previously by our group and other researchers on the combination of xCT targeting and chemotherapy. This paragraph has been added in the conclusion, Line n° 603-610.

Reviewer 2 Report

The authors describe VLP-based vaccine for metastatic breast cancer. The manuscript is well written and conceptually sounds

Following points need to be considered:

  1. I suggest the authors to point out the novelty and innovation potential of their study. This will help for a broader reach of the study
  2. I also recommend to include the state of the art in the introduction and limitations associated with it.
  3. Why the authors have represented the data in figure 2 in terms of OD and not log titers. I suggest to change the representation of the data as those values are more relatable
  4. Similar to figure 2, the figure 6A also has the representation is OD (optical denisity), is there a specific reason the authors have choosen to represent in OD

Author Response

Response to Reviewer#2 comments

  1. I suggest the authors to point out the novelty and innovation potential of their study. This will help for a broader reach of the study

RESPONSE 1:

We agree with the Reviewer and we have better described the innovation of our study in the Introduction section, Line n°80-84.

  1. I also recommend to include the state of the art in the introduction and limitations associated with it.

RESPONSE 2:

We thank the Reviewer for this suggestion. We have now added a short description of the therapeutic scenario of metastatic breast cancer in the Introduction, Lines n° 37-41.

  1. Why the authors have represented the data in figure 2 in terms of OD and not log titers. I suggest to change the representation of the data as those values are more relatable. Similar to figure 2, the figure 6A also has the representation is OD (optical density), is there a specific reason the authors have choosen to represent in OD.

RESPONSE 3:

We performed the ELISA of fig. 2A and 2B as End Point Titer ELISA and using sera from each single mouse. However, the other ELISA shown in the paper were performed using sera from each single mouse only at the indicated dilution, that is the same used for the in vitro experiments. In order to express all the ELISA results in the same way, we decided to report OD values.

We think that the presentation of the ELISA in terms of OD is not unusual. Several labs present it in the same manner on their publications. In our case since we used the same concentration of the antigen (ECD3 peptide), the OD gives a good representation of the antibody binding and its distribution among the group of mice. Moreover, this renders the data easily comparable to those reported in our previous xCT publication (Bolli E et al., Oncoimmunology 2018; figure 2A and 2B).